# TabMT: Generating Tabular data with Masked Transformers

**Manbir S. Gulati**
AI Accelerator
Leidos Inc
Manbir.S.Gulati@leidos.com

**Paul F. Roysdon**
AI Accelerator
Leidos Inc
Paul.Roysdon@leidos.com

## Abstract

Autoregressive and Masked Transformers are incredibly effective as generative models and classifiers. While these models are most prevalent in NLP, they also exhibit strong performance in other domains, such as vision. This work contributes to the exploration of transformer-based models in synthetic data generation for diverse application domains. In this paper, we present TabMT, a novel Masked Transformer design for generating synthetic tabular data. TabMT effectively addresses the unique challenges posed by heterogeneous data fields and is natively able to handle missing data. Our design leverages improved masking techniques to allow for generation and demonstrates state-of-the-art performance from extremely small to extremely large tabular datasets. We evaluate TabMT for privacy-focused applications and find that it is able to generate high quality data with superior privacy tradeoffs.

## 1 Introduction

Generative models have attracted significant attention in the field of deep learning due to their ability to synthesize high-quality data and learn the underlying structure of complex datasets. Such models have been successfully applied to various data types including images [18], text[8], and tabular data [12]. This work concentrates on tabular data, which is prevalent in numerous fields like healthcare, finance, and social sciences. The heterogeneous nature of tabular data, characterized by its diverse data types, distributions, and relationships, presents distinct challenges not present in other domains.

The development of effective synthetic tabular data generators is crucial for numerous reasons including: privacy preservation, data augmentation, model interpretability, and anomaly detection. Prior work in this domain has produced a myriad of generative models, including Generative Adversarial Networks (GANs)[28][25], Variational Autoencoders (VAEs)[25], Autoregressive Transformer[22] [1], and Diffusion models[12]. Although these existing models strive to address the challenges associated with tabular data generation, there is still room for exploration and improvement. Specifically, we demonstrate improvements in robustness, scalability, privacy preservation, and handling of missing data.

Transformers [23], originally designed for natural language processing (NLP) tasks, have lead to significant advancements in a variety of applications. Their powerful capacity for modeling complex dependencies and

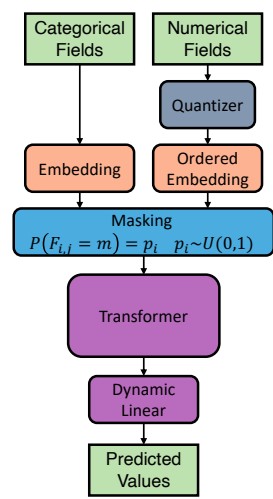

Figure 1: Diagram of TabMT. $m$ is the mask token, $p_i$ is the masking probability of the $i^{th}$ row

37th Conference on Neural Information Processing Systems (NeurIPS 2023).

generalizing across applications has spurred researchers to extend transformers to other data types, such as images[11][4] and audio[17].

In this paper, we investigate transformers as synthetic tabular data generators, further expanding their cross-domain applicability. We specifically examine Masked Transformers (MT), originally developed to produce strong text embeddings [5], and successfully generalized across many domains[9][3][10]. We explore their utility as tabular data generators. We show Masked Transformers make robust and scalable tabular generators, achieving state-of-the-art performance across a wide array of datasets.

Our key contributions are as follows:

1. We propose TabMT, see Figure 1, a simple but effective Masked Transformer design for generating tabular data, that is general enough to work across many tasks and scenarios.

2. We provide a comprehensive evaluation of TabMT, demonstrating state-of-the-art performance when compared to existing generative model families, including GANs, VAEs, Autoregressive Transformers, and Diffusion models. We also showcase its scalability from very small to very large tabular datasets.

3. We highlight the applicability of our model in privacy-focused applications, illustrating TabMTs ability to arbitrarily trade-off privacy and quality through temperature scaling. Furthermore, our model's masking procedure enables it to effectively handle missing data, thereby increasing privacy and model applicability in real-world use cases.

## 2  Related Work

**Language modeling:**    Transformer[23] based Language Modeling[5][8] learns Human language via token prediction. Given a context of either previous tokens in time, or a random subset tokens within a window, the model is tasked with predicting the remaining tokens. Using either an autoregressive model or masked model, respectively. Both these paradigms have shown success across a wide range of tasks and domains.[9][4][17][26][3] Masked Language models are traditionally used for their embeddings, although some papers have explored masked generation of content[3]. Our work builds on Masked Training and demonstrates its effectiveness for modeling and generating tabular data.

**Deep Tabular Generators:**    Deep learning models are increasingly utilized for generating synthetic tabular data. Synthetic data is particularly important for tabular data as it is often subject to privacy requirements. Additionally, tabular datasets are often hard to acquire, and usually smaller than datasets in other domains. These conditions increase the importance and challenges associated with generating tabular data. Deep Tabular Generators have been constructed using essentially all Deep Generative model families Autoencoders[25], GANs[25][28][27][7], and AR transformers [22] [1] [16]. Our work uses Masked Transformers and demonstrates superior performance when compared to prior deep tabular generators.

**Netflow Generators:**    Netflow data is a specific type of tabular data that captures network communication events and is commonly used for network traffic analysis, intrusion detection, and cybersecurity applications. Netflow datasets are typically extremely large, with complex rules between fields, and a high number of possible values per field. Generating realistic synthetic netflow data is crucial for developing and testing network monitoring tools and security algorithms. There have been several models developed specifically for netflow generation [21] [24] [15]. We demonstrate that our general tabular data generator handily outperforms domain specific models.

## 3  Method

TabMTs structure is particularly well suited for generating tabular data, for a number of reasons.

1. TabMT accounts for patterns bidirectionally between fields. Tabular data lacks ordering, meaning bidirectional learning likely produces better understanding and embeddings within the model.

2. A "prompt" to a tabular generator is not likely sequential. TabMT's masking procedure allows for arbitrary prompts to the model during generation. This is unique as most other generators have very limited conditioning capabilities.

3. Missing data is far more common in tabular data than in other domains. TabMT is able to learn missing values by setting their masking probability to 1. Other generators require that we impute data separately before we can generate high-quality cleaned samples.

These structural advantages come from TabMT's novel masked generation procedure.



Figure 2: A row of data being sampled from TabMT. Fields are sampled in a random order and field values are sampled according to the predicted distributed.

Below we outline how we construct TabMT from the original masked training procedure outlined in BERT[5] and the justifications behind our design choices. We also outline the specific changes we make to allow for heterogeneous data types. A naive adaptation of BERT's masking procedure would look as follows. Given an $n$ by $l$ dataset $\mathbf{F}$ of categorical and numerical features, for each row $\mathbf{F}_i$, the transformer is provided with a set of unmasked fields $\mathbf{F}_i^u$ and a set of masked fields $\mathbf{F}_i^m$. Each field in the masked set $\mathbf{F}_i^m$ has its value replaced with a mask token. The model is then tasked with predicting the original value for all masked tokens. The row $\mathbf{F}_i$ is partitioned into the unmasked and masked sets by conducting a Bernoulli trial on each field, $\mathbf{F}_{i,j}$, such that $P(\mathbf{F}_{i,j} \in \mathbf{F}_i^m) = 0.15$.

The BERT masking procedure produces a strong embedding model, but not a strong generator. To create a strong generative model we make two key changes: sample our masking probability from a uniform distribution and predict masked values in a random order during generation. To understand why these changes are effective, we can look at the distribution of masked sets. As a result of the repeated Bernoulli trials during masking, the size of the masked set for each row $|\mathbf{F}_i^m|$ will follow a Binomial distribution. However, when generating data one field $\mathbf{F}_{i,j}$ at a time, the model will inference on masked subset sizes $(0, \ldots, l-1)$, once each. We would like the training distribution of $|\mathbf{F}_i^m|$ to be uniform, matching the uniform distribution encountered when generating data. With a fixed masking probability $P(\mathbf{F}_{i,j} \in \mathbf{F}_i^m) = p_m$ we will instead encounter a Binomial distribution centered around $p_m \cdot l$. However, if we sample our masking probability $p_m$ for each row $\mathbf{F}_i$ such that $P(p_m = p) \sim U(0, 1)$, we will train uniformly across subset sizes:

$$P(|\mathbf{F}_i^m| = k) = \int_0^1 \binom{l}{k} p^k (1-p)^{l-k} dp = \frac{l!}{k!(l-k)!} \frac{k!(l-k)!}{(l+1)!} = \frac{1}{l+1}. \tag{1}$$

Fixing this train and inference mismatch is critical to forming a strong generator. A traditional autoregressive generator would generate fields $(\mathbf{F}_{i,0}, \ldots, \mathbf{F}_{i,l-1})$, sequentially. However, tabular data, unlike language, does not have an inherent ordering. Generating fields in a fixed order introduces another mismatch between training and inference. During training $\mathbf{F}_i^m$ will take on the distribution

$$P(\mathbf{F}_i^m = s) = \frac{1}{\binom{l}{|s|} \cdot l}. \tag{2}$$

When generating in a fixed order, the model will infer across $l$ distinct subsets and no others. However, if we instead infer in a random order, then at generation step $0 \leq t < l$, the distribution of $\mathbf{F}_i^m$ will be given by

$$P(\mathbf{F}_i^m = s) = \frac{t! \cdot (l-t)!}{l!} = \frac{1}{\binom{l}{t}}. \tag{3}$$

Since we encounter each $t$ exactly once, this overall distribution is identical to the masking distribution encountered during training, fixing the discrepancy caused by generating fields in a fixed order.

A transformer model will typically have an input embedding matrix $\mathbf{E} \in \mathbb{R}^{k \times d}$, where $k$ is the number of unique input tokens and $d$ is the transformer width. Because tabular data is heterogeneous, we instead construct $l$ embedding matrices, one for each field. Each embedding matrix will have a different number of unique tokens $k$.

For categorical fields we use a standard embedding matrix initialized with a normal distribution. For each continuous field we construct an ordered embedding $\mathbf{O} \in \mathbb{R}^{k \times d}$ from its unordered embedding matrix $\mathbf{E}$ and two endpoint vectors $\mathbf{l}, \mathbf{h} \in \mathbb{R}^d$.

To construct each ordered embedding matrix $\mathbf{O}$, we first we quantize the values of the continuous field. Our default quantizer is K-Means. We consider the maximum number of clusters a hyper-parameter. Let $\mathbf{v} \in \mathbb{R}^k$ be the vector of ordered cluster centers. We construct an vector of ratios $\mathbf{r} \in \mathbb{R}^k$ using min max normalization such that

$$\mathbf{r}_i = \frac{\mathbf{v}_i - \min(\mathbf{v})}{\max(\mathbf{v}) - \min(\mathbf{v})}. \tag{4}$$

We use the ratio vector $\mathbf{r}$ to construct each ordered embedding in $\mathbf{O}$:

$$\mathbf{O}_i = \mathbf{E}_i + \mathbf{r}_i \cdot \mathbf{l} + (1 - \mathbf{r}_i) \cdot \mathbf{h}. \tag{5}$$

This structure allows the transformer to both take advantage of the ordering of the properties and add unordered embedding information to each cluster. The unordered embeddings are useful in attention, multi-modal distributions, and encoding semantic separation between close values. We use this same structure to construct a dynamic linear layer at the output during prediction. This can be converted to a traditional linear layer once the model is trained.

Relying too heavily on the unordered embeddings might negate the benefit of our ordered embedding, as information isn't effectively shared between close values. To address this, we bias TabMT to rely on the ordering of embeddings. For continuous fields, we zero-init the unordered embedding matrix $\mathbf{E}$. Whereas, the endpoint vectors $\mathbf{l}$ and $\mathbf{h}$ use a normal distribution with 0.05 standard deviation. Because entries in matrix $\mathbf{O}$ are not independent of each other, to sharpen the output distribution, the network must either rely on matrix $\mathbf{E}$ or increase magnitude of the endpoint vectors. This can reduce use of the priors encoded by $\mathbf{O}$ or cause instability. To combat this, we include a learned temperature which can sharpen the predicted distribution using a single parameter per field instead. Each field's predicted distribution $\hat{\mathbf{y}} \in \mathbb{R}^k$ is given by

$$\hat{\mathbf{y}} = \frac{e^{\mathbf{z}/(\tau_l \cdot \tau_u)}}{\sum_j e^{\mathbf{z}/(\tau_l \cdot \tau_u)}}, \tag{6}$$

where $\mathbf{z} \in \mathbb{R}^k$ is a vector of logits, $\tau_l$ is the learned temperature, and $\tau_u$ is the user-defined temperature. See Figure 1 for an overall diagram of these components. Figure 2 shows the generation of a single sample. Detailed pseudocode is available in the Appendix.

## 4 Evaluation

In this section, we present a comprehensive evaluation of TabMT's effectiveness across an extensive range of tabular datasets. Our analysis involves a thorough comparison with state-of-the-art approaches, encompassing nearly all generative model families. To ensure a robust assessment, we evaluate across several dimensions and metrics.

**Datasets:** For our data quality and privacy experiments we use the same list of datasets and data splits as TabDDPM[12]. These 15 datasets range in size from $\sim 400$ samples to $\sim 150,000$ samples. They contain continuous, categorical, and integer features. The datasets range from 6 to 50 columns. For our scaling experiments we use the CIDDS-001[20] dataset, which consists of Netflow traffic from a simulated small business network. A Netflow consists of 12 attributes, which we post-process into 16 attributes; see the Appendix for more details. This dataset is extremely large with over 30 million rows and field cardinalities in the tens of thousands. Other datasets listed all have cardinalities below 50. Unlike our other benchmarks, we purposely do not quantize the continuous variables here to further test the scaling of our model. In other words, every unique value is treated as a separate category in our prediction process.

**Prior Methods:** We select four techniques to compare against, one from each each major family of deep generative models.

- **TVAE**[25] is one of the first deep tabular generation papers introducing two models, a GAN and a VAE. We compare against their VAE because it is the strongest VAE tabular generator we are aware of.

- **CTABGAN+**[28], at the time of writing, is the state-of-the-art for GAN-based tabular synthesis.

- **TabDDPM**[12] adapts diffusion models to tabular data, with the strongest results of all prior work.

- **RealTabFormer**[22] is a recent work on adapting autoregressive transformers to tabular and relational data. This method is most similar to our technique, however, they use an autoregressive transformer which demonstrates worse results than our masked transformer.

## 4.1 Data Quality

We use the CatBoost variant of ML Efficiency (MLE)[27][12] for evaluating the quality of our synthetic data. This metric trains a CatBoost[6] model on the synthetic data instead of a weak ensemble. The CatBoost model is able to detect subtle patterns in the data, that weak classifiers cannot. This is a holistic metric that accounts for both diversity and quality of samples. For a fair comparison we use the the standard hyper-parameter tuning budget of 50 trials[28][25]. Our full search space is provided in the Appendix. Each data quality experiment was conducted using a single A10 GPU each. For evaluation, we generate scores and standard deviations[1] on the test set, training a CatBoost model 10 times on 5 samples of synthetic data.

Table 1: MLE and standard deviations across techniques. The highest MLE score for each dataset is highlighted in **bold**. $F_1$ is used for classification. $R^2$ is used for regression tasks. *: the source paper [22] cites a $\sim 3\%$ higher test accuracy using the real train set over what other papers achieve, likely because they used a different split or version of this dataset.

| DS | TVAE | CTabGAN+ | RealTab. | TabDDPM | TabMT | Real |
|---|---|---|---|---|---|---|
| AB | 0.433±0.008 | 0.467±0.004 | 0.504±0.011 | **0.550±0.010** | 0.535±0.004 | 0.556±0.004 |
| AD | 0.781±0.002 | 0.772±0.003 | 0.811±0.002 | 0.795±0.001 | **0.814±0.001** | 0.815±0.002 |
| BU | 0.864±0.005 | 0.884±0.005 | 0.928±0.003* | 0.906±0.003 | **0.908±0.002** | 0.906±0.002 |
| CA | 0.752±0.001 | 0.525±0.004 | 0.808±0.003 | 0.836±0.002 | **0.838±0.002** | 0.857±0.001 |
| CAR | 0.717±0.001 | 0.733±0.001 | - | 0.737±0.001 | **0.738±0.001** | 0.738±0.001 |
| CH | 0.732±0.006 | 0.702±0.012 | - | **0.755±0.006** | 0.741±0.005 | 0.740±0.009 |
| DI | 0.714±0.039 | 0.734±0.020 | 0.732±0.027 | 0.740±0.020 | **0.769±0.018** | 0.785±0.013 |
| FB | 0.685±0.003 | 0.509±0.011 | 0.771±0.004 | 0.713±0.002 | **0.798±0.002** | 0.837±0.001 |
| GE | 0.434±0.006 | 0.406±0.009 | - | 0.597±0.006 | **0.605±0.008** | 0.636±0.007 |
| HI | 0.638±0.003 | 0.664±0.002 | - | 0.722±0.001 | **0.727±0.001** | 0.724±0.001 |
| HO | 0.493±0.006 | 0.504±0.005 | - | **0.677±0.010** | 0.619±0.004 | 0.662±0.003 |
| IN | 0.784±0.010 | 0.797±0.005 | - | 0.809±0.002 | **0.811±0.003** | 0.814±0.001 |
| KI | 0.824±0.003 | 0.444±0.014 | - | 0.833±0.014 | **0.876±0.011** | 0.907±0.002 |
| MI | 0.912±0.001 | 0.892±0.002 | - | 0.936±0.001 | **0.938±0.001** | 0.934±0.000 |
| WI | 0.501±0.012 | 0.798±0.021 | - | **0.904±0.009** | 0.881±0.009 | 0.898±0.006 |

MLE scores are presented in Table 1; note that we match or exceed state-of-the-art on 11 of 15 datasets. To gain a qualitative understanding of data quality we visualize the distribution of correlation errors; see Figure 3. To calculate the correlation errors, we first compute the correlation $\mathbf{r}_{i,j}$ between each pair of fields $(i, j)$. To compute correlations involving categorical columns, we convert them to one-hot vectors. We then compute the correlation between columns on the synthetic data $\hat{\mathbf{r}}_{i,j}$. The correlation error is the absolute difference between these two values $|\mathbf{r}_{i,j} - \hat{\mathbf{r}}_{i,j}|$. These errors should approach zero because we expect the correlations between columns in the synthetic data to be the same as those in the real data. Erroneous and missing correlations appear as non-zero values in the histograms of Figure 3.

---

[1]For comparison, we use the same error reporting method as prior work, however, we plan to present a thorough coverage of more accurate error estimation for generative models in future work.

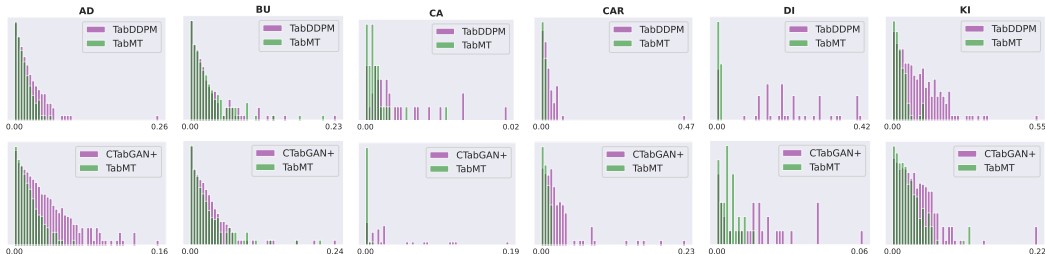

Figure 3: A comparison of Correlation Error Histograms between TabMT (Green) vs TabDDPM (Purple) and TabMT (Green) vs CTabGAN+ (Purple). A good generator should have correlation errors distributed close to zero (the left of the plot). We can see TabMT's correlation errors are consistently distributed closer to zero than either TabDDPM or CTabGAN+.

## 4.2 Privacy and Sample Novelty

Maintaining privacy of the original data is a key application for synthetic tabular data. Machine learning is increasingly being used across a wide range of areas to produce valuable insights. Simultaneously, there is a rapid rise in both regulation and privacy concerns that need to be addressed. In Section 4.1 we demonstrated that data produced by TabMT is high enough quality to produce strong classifiers. Now we evaluate our model for privacy. This evaluation complements our quality evaluation and verifies that our model is generating novel data. Novelty means data is not substantially similar to samples encountered during training. A high-quality non-private model can trivially be formed by directly reproducing the training set exactly. None of this data is novel, but it is high quality. By ensuring our model is both private and high quality, we verify that our model has learned the distribution of the data, and not simply memorized the training set. Memorization is a larger issue in tabular data due to smaller dataset sizes and increased privacy concerns.

To evaluate privacy and novelty we adopt the median Distance to the Closest Record (DCR) score. To calculate the DCR of a synthetic sample, we find the nearest neighbor in the real training set by Euclidean distance. We report the median of this distance across our synthetic samples. Data with higher DCR scores will be more private and more novel. There is an inherent trade off between privacy and quality. Higher quality samples will tend to be closer to points in the training set and vice versa.

While models such as CTabGAN+[28] and TabDDPM[12] have a fixed trade-off between privacy and quality after training. TabMT can trade-off between quality and privacy using temperature scaling. By walking along the Pareto curve of our model, using temperature scaling, we can controllably tune the privacy and novelty of our generated data per application. By increasing a field's temperature, its generated values become more novel and private, but they are also less faithful to the underlying data distribution. The trade off between the quality and privacy here form a Pareto front for TabMT on each dataset.

We use a separate temperature for each field and perform a search to estimate the Pareto front. Each search was conducted using a single A10 GPU each. Search details are available in the Appendix. In Table 2, we compare TabMT's DCR and corresponding MLE scores to that of TabDDPM. We are always able to attain a higher DCR score, and in most cases a higher MLE score as well. This falls in line with recent results in other domains showing diffusion models are less private than other generative models[2]. A comparison with CTabGAN+ is available in the Appendix, compared to CTabGAN+ we obtain both higher privacy and MLE scores in all tested cases. Figure 4 shows the Pareto fronts of TabMT across several datasets.

## 4.3 Missing Data

Real world data is often missing many values that make training difficult. When a row has a missing value we must either drop the row, or find a method to impute the missing value. Other techniques such as the RealTabformer[22] or TabDDPM[12] cannot natively handle real world missing data, and must either use a different imputation technique or drop the corresponding rows. Our masking procedure allows TabMT to natively handle arbitrary missing data. To demonstrate this, we randomly drop 25% of values from the dataset, ensuring nearly every row is permanently missing data. Nevertheless, our

Table 2: DCR score comparison between TabDDPM and TabMT. Corresponding MLE scores are in parentheses.

| DS | TabDDPM | **TabMT** |
|----|---------|-----------|
| AB | 0.050(0.550) | **0.249**(0.533) |
| AD | 0.104(0.795) | **1.01**(0.811) |
| BU | 0.143(0.906) | **0.165**(0.908) |
| CA | 0.041(0.836) | **0.117**(0.832) |
| CAR | 0.012(0.737) | **0.041**(0.737) |
| CH | 0.157(0.755) | **0.281**(0.758) |
| DI | 0.204(0.740) | **0.243**(0.740) |
| FB | 0.112(0.713) | **0.252**(0.787) |

| DS | TabDDPM | **TabMT** |
|----|---------|-----------|
| GE | 0.059(0.597) | **0.234**(0.599) |
| HI | 0.449(0.722) | **0.483**(0.727) |
| HO | 0.086(0.677) | **0.151**(0.607) |
| IN | 0.041(0.809) | **0.061**(0.816) |
| KI | 0.189(0.833) | **0.335**(0.868) |
| MI | 0.022(0.936) | **0.026**(0.936) |
| WI | 0.016(0.904) | **0.063**(0.881) |

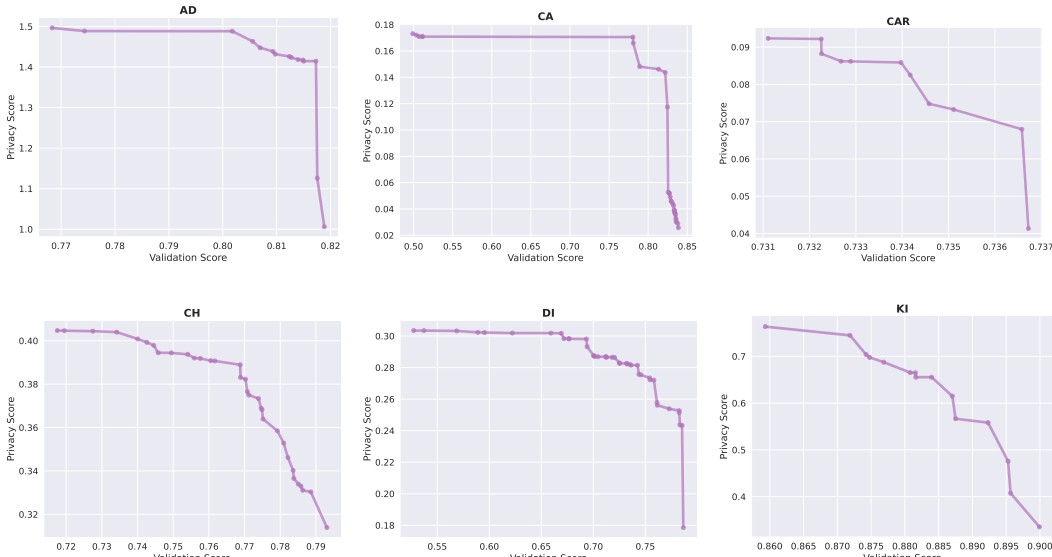

Figure 4: Pareto Fronts of TabMT balancing the tradeoff between privacy (DCR) and data quality (Validation Score). While some datasets have a smooth transition the temperature changes, others have a sharp drop-off.

model is still able to train, producing synthetic rows with no missing values in them. This facilitates training on real world data.Table 3 shows our accuracy when training with missing data.

Table 3: MLE of TabMT when training with 25% of values missing. Delta represents the difference in MLE from training with no missing values.

| DS | MLE | Delta |
|----|-----|-------|
| AD | 0.813 | -0.001 |
| KI | 0.868 | -0.008 |

Additionally, our model can be arbitrarily conditioned to produce any subset of the data distribution at no additional cost, allowing us to more effectively augment underrepresented portions of data. Prior art is largely incapable of conditioning when producing outputs.

## 4.4 Scaling

In this section, we examine the performance of our model when scaling to very large datasets. We use the CIDDS-001 dataset[20] as our benchmark dataset. We do not use anomalous traffic from the dataset, and randomly select 5% of the dataset as the validation set for reporting results. The results

in this section, together with those in Section 4.1, demonstrate that TabMT is both sample-efficient enough to learn with just a few hundred samples, while remaining general enough to scale to over thirty million samples. We train three model sizes on this dataset and compute metrics on the resulting samples. The model topologies are outlined in Table 4. Each model was trained on a single A10 GPU with the exception of TabMT-L which was trained using 4 V100s. We use the AdamW[14] optimizer with a learning rate of 0.002 and weight decay of 0.01, a batch size of 2048 and a cosine annealing learning rate schedule for 350,000 training steps and 10000 warm-up steps.

Table 4: Model topologies used in scaling experiments. The large model sizes here demonstrate we can scale well in terms of model size and dataset size.

| Model | Width | Depth | Heads |
|-------|-------|-------|-------|
| TabMT-S | 64 | 12 | 4 |
| TabMT-M | 384 | 12 | 8 |
| TabMT-L | 576 | 24 | 12 |

Because ML Efficiency and DCR are very costly to compute on a dataset of this scale, we instead adapt Precision and Recall[13] to the tabular domain. The original definitions of these metrics[13] rely on vision models to produce the embeddings used. We find our masking procedure produces strong embeddings for each sample, so we use the embeddings produced by TabMT-S. Specifically, we average the embeddings across the fields to produce 64 dimensional for each flow. We used a fixed neighborhood size of $k = 3$. We include an additional diversity metric defined as the average set coverage across all properties of the generated data. Results in Table 5 demonstrate strong scaling and performance across model sizes. We can see Precision and Recall are both very close to that of the validation set. Sample diversity and quality both scale as model size increases.

Table 5: Precision, Recall, and Diversity metrics for all tested models. TabMT outperforms NFGAN[21] on all metrics

| Source | Validation Set | NFGAN | TabMT-S | TabMT-M | TabMT-L |
|--------|----------------|-------|---------|---------|---------|
| Precision(%) | 90.42 | 77.64 | 82.58 | 85.77 | 88.10 |
| Recall(%) | 90.58 | 63.32 | 91.88 | 91.82 | 91.12 |
| Diversity(%) | 100.0 | 46.97 | 89.81 | 90.56 | 99.43 |

We compare against the prior state-of-the-art NetflowGAN, or NFGAN[21]. This GAN was tuned specifically for this dataset. It is trained in two phases. First IP2Vec[19] is trained to produce Netflow embeddings. These embeddings are then used as targets for the generator during GAN training. Results from NFGAN are shown in Table 5. We can see that NFGAN obtains reasonably high precision, but poor recall and diversity. This is because the model suffers from mode collapse, producing samples in only a small portion of the full distribution.

Netflow has both correlations between the fields and complex invariants between fields. We can measure the violation rate of these invariants to understand how well our model is detecting patterns within the data. We measure against seven invariants proposed in [21]. As shown in Table 6 TabMT produces substantially more diverse data, while achieving a median 20x improvement in violation probability over NFGAN.

# 5 Limitations and Future Work

TabMT presents strong results even on large datasets, but our Transformer backbone means TabMT is slower than more lightweight methods built around small MLPs, or GANs which can produce a row in a single inference. Searching temperatures also adds time if optimal privacy is needed. Additionally, we must quantize continuous fields, while we outperform methods which do not quantize fields, this could pose issues in some applications. Future work might examine learning across tabular datasets, alternative masking procedures and networks to improve speed, or integration with diffusion models to better tackle continuous fields.

Table 6: Error rates on netflow invariant tests. *: because we construct embeddings per field, our model cannot violate check 5. These tests check structural rules reflected in Netflow, such as the fact that two public ip addresses cannot communicate to each other.

|  | NFGAN | TabMT-S | TabMT-M | TabMT-L |
|---|---|---|---|---|
| TCP Flags | 2.33e-03 | 4.63e-04 | 2.16e-04 | **6.54e-06** |
| Private IPs | 2.00e-04 | 7.25e-05 | 2.67e-05 | **1.14e-05** |
| TCP Port | 3.00e-04 | 9.32e-05 | 3.94e-05 | **1.47e-05** |
| DNS | 1.60e-03 | 2.34e-03 | 1.56e-03 | **2.05e-04** |
| Valid Values* | 2.00e-03 | **0.00e-00** | **0.00e-00** | **0.00e-00** |
| NetBios | 7.43e-01 | 4.73e-03 | 4.64e-03 | **1.27e-03** |
| Packet Ratios | 5.10e-03 | 2.28e-03 | 1.79e-03 | **1.16e-03** |

**Broader Impact** Synthetic data generation allows for privacy preservation, protecting sensitive data while still enabling data analysis. High Quality synthetic data may ease the pressure to resort to unethical methods of collection such as relying on underpaid labor. With this in mind, trading off data for additional compute does mean that the additional compute will contribute to increased $CO_2$ emissions. Additionally, synthetic data carries the risk of misuse, such as the potential for manipulating results or research findings with fabricated data. All experiments were conducted using cloud A10 or V100 GPUs. For algorithm design and experiment result generation roughly 410 GPU days of compute were used.

## 6 Conclusion

In this paper, we outlined a novel Masked Transformer design and training procedure, TabMT, for generating synthetic tabular data. Through a comprehensive series of benchmarks we demonstrate that our model achieves state-of-the-art generation quality. This quality is verified at scales that are orders of magnitude larger than prior work and with missing data present. Our model achieves superior privacy and is able to easily trade off between privacy and quality. Our model is a substantial advancement compared to previous work, due to its scalability, missing data robustness, privacy-preserving generation, and superior data quality.

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
