# OpenReview forum: "TabMT: Generating tabular data with masked transformers"
_NeurIPS.cc/2023/Conference — NeurIPS 2023 poster_

### Official Review · Reviewer_EYXT · 2023-06-19

**Soundness:** 2 fair
**Presentation:** 3 good
**Contribution:** 2 fair
**Rating:** 4
**Confidence:** 2

**Summary:**

This paper introduces TabMT, a new Masked Transformer architecture designed for generating synthetic tabular data. While Transformers are predominantly used in natural language processing (NLP), TabMT demonstrates their effectiveness in dealing with heterogeneous data fields, such as images and tables, and efficiently manages missing data. The authors propose to sample the masking probability from a uniform distribution and predict masked values in random order during generation. By employing advanced masking techniques, TabMT can generate synthetic data with high performance across a wide range of dataset sizes. Moreover, the model proves valuable in privacy-sensitive applications, as it is capable of producing high-quality data while adhering to privacy restrictions.

**Strengths:**

The motivation is easy to understand and the problem is important but less explored than common domains. It is not straightforward to apply existing techniques on images and texts to the tabular data.
The experimental designs are comprehensive, including several sections assessing data quality, Privacy and Sample Novelty, Missing Data, and Scaling, which are solid.

**Weaknesses:**

The motivations for designs are not very clearly written. Please consider reorganizing the structure, adding highlighted paragraphs, and moving line 129-138 to the beginning of the method or merging them with the methodology introduction texts.

The introduction of temperature scaling is not well-motivated, is it for privacy purposes and why will it work?

The introduction of privacy tabular data generation is not detailed. The conclusion claims that "the model is able to function under arbitrary privacy budgets." but I even didn't see how the budgets are computed and its defitions, motivations etc. Same for the definition of data "novelty". And the questions of "why many designs in TabMT can help those aspects are not well explained".



**Questions:**

The work can benefit from discussing and comparing the work of causal inference using transformers, which is a particular kind of missing data generation problem where privacy protection is also pressing, e.g. [1], [2].

[1] Exploring Transformer Backbones for Heterogeneous Treatment Effect Estimation
[2] Differentially Private Synthetic Control

**Limitations:**

See the weaknesses part

---

> ### Author Rebuttal · Authors · 2023-08-08
>
> Hi Reviewer EYXT, Thank you for the time you took to read our paper and write your review.
>
> **W:** The motivations for designs are not very clearly written. Please consider reorganizing the structure, adding highlighted paragraphs, and moving line 129-138 to the beginning of the method or merging them with the methodology introduction texts.
>
> **R:** We can reorganize these things and add bolding to make things clearer. 129-138 would likely work better at the top of this section.
>
> **W:** The introduction of temperature scaling is not well-motivated, is it for privacy purposes and why will it work?
>
> **R:** Temperature Scaling is often used when generating data from transformers to increase diversity or improve generation quality (see the references below). We can add a few more sentences explaining our learned temperature. Our learned temperature allows the network to sharpen the output logits across the embeddings. Allowing sharpening through temperature gives the model a more effective method of doing so without relying on the unordered embeddings or magnitude changes. This is important in our ordered embeddings since they are interpolations, instead of each embedding being independent of each other. We have confirmed the learned temperature acts in this way as all learned temperatures have averages well below 1 (sharpening), and nearly all end up below their initialized value.
>
> **References:**
>
> Temperature use during generation to improve generation quality or diversity:
>
> https://arxiv.org/pdf/1904.09751.pdf
>
> https://transformer.huggingface.co/doc/gpt2-large (text generation demo with temperature)
>
> Temperature during Learning:
>
> https://arxiv.org/pdf/2002.05709.pdf (hyperparameter which helps sharpen the logits)
>
> https://arxiv.org/pdf/2103.00020.pdf (learned temperature instead of hyperparamter)
>
> **W:** The introduction of privacy tabular data generation is not detailed. The conclusion claims that "the model is able to function under arbitrary privacy budgets." but I even didn't see how the budgets are computed and its defitions, motivations etc. Same for the definition of data "novelty". And the questions of "why many designs in TabMT can help those aspects are not well explained".
>
> **R:** Lines 189-203 help address this within the paper. Here we define the Distance to Closest Record metric (DCR), which has also been used in prior work, but allow us to explain things further. Perhaps instead of saying “arbitrary privacy budgets”, we should say “produce data with arbitrary privacy scores”. By budget we mean a privacy score (DCR) threshold. The model is able to operate under arbitrary privacy budgets since we are able to walk the Privacy v. Quality Pareto curves of our model. **Most previous works do not have the ability to change the privacy scores of their generated data after training**. Figure 4 is a good illustration of this concept. Imagine we would like our DCR to be above some number for our synthetic data (the threshold is our budget), we can change the temperatures of our model to achieve this DCR, no matter what the DCR is, up to the point of generating random data. Other models will have a fixed DCR after training and if that falls below what you desire, then that model becomes unusable for your application. Our model is tunable in this respect in a way most others aren’t and can function across all reasonable thresholds, because the temperature can be tuned in order to achieve it. We also show that at high quality scores our model also achieves higher privacy than models of similar quality. This demonstrates temperature works well to control this tradeoff in our model.
>
> DCR is also a good measure of novelty in addition to privacy because as samples get further away in terms of distance from all points seen during training, we can expect them to generally be more novel. Here’s another quote from the paper related to this.
>
> *By ensuring our model is both private and high quality, we verify that our model has learned the intrinsic structure of the data, and not simply memorized it.*
>
> This is what we mean by novelty. If our synthetic dataset is far in terms of distance from the training set, the model could not have simply memorized it. Our model has SoTA privacy and quality scores, meaning it produces novel data which is also high quality.
>
> Previous Work in this field using DCR for these purposes:
>
> RealTabFormer: https://arxiv.org/abs/2302.02041
>
> TabDDPM: https://arxiv.org/pdf/2209.15421v1.pdf
>
> CTabGAN: https://arxiv.org/pdf/2102.08369.pdf
>
> Nearest neighbors are commonly examined from the training set, even in other domains, to see if generative models are producing truly novel data and are not memorizing or overfitting (e.g. BigGAN: https://arxiv.org/pdf/1809.11096.pdf)
>
> **Q:** The work can benefit from discussing and comparing the work of causal inference using transformers, which is a particular kind of missing data generation problem where privacy protection is also pressing, e.g. [1], [2].
>
> **A:** We can add a discussion of this work within the related work section to talk about the challenges these techniques solve and the different approach they take for it.
>
> Our model achieves state of the art generation quality and privacy while having the ability to tradeoff between these arbitrarily, natively learn with missing data present, and condition on arbitrary subsets of the distribution during generation. We are not aware of any prior art which is able to do these things within a singular model, certainly not while also achieving state of the art generation quality. We achieve these results while using an innovative masked generator architecture, and thoroughly validate our architecture up to tens of millions of rows.
>
> Thank you very much for the time you spent reviewing and reading our paper. We hope these comments can help clarify some of the details of our paper, and why we believe it to be a worthwhile and novel contribution to Tabular Data Synthesis.

---

### Official Review · Reviewer_aH7d · 2023-07-06

**Soundness:** 3 good
**Presentation:** 3 good
**Contribution:** 3 good
**Rating:** 7
**Confidence:** 3

**Summary:**

The paper proposes a masked transformer model that can be used to generate tabular data. The authors propose modifications in the masking strategy in the transformer, in order to make it more effective for generating data. Empirically, the proposed generator improves performance in various datasets compared to other baselines. Moreover, the paper presents cases of privacy-focused applications, presence of missing values, and large datasets to further depict the usability of the proposed method in real-world situations.

**Strengths:**

- The paper is well-written and very clear to its points. The authors define the problem with the masked transformer for generating tabular data, and presents an effective method to overcome the difficulty. The empirical results are well presented, and seems promising.

**Weaknesses:**

As noted in the paper, the major weakness would be the resource it takes to train the model. For general users, a pre-trained model with learning across multiple tables might be beneficial.

**Questions:**

- Is it possible for the model to generate new categories (or numbers) not present in the train set?


**Limitations:**

The presented work might not be accessible to regular users, given the amount of resource required to train the model.

---

> ### Author Rebuttal · Authors · 2023-08-08
>
> Hi Reviewer aH7d, thank you for your rating and the time you took to carefully read and review our paper.
>
> **W:**  ”As noted in the paper, the major weakness would be the resource it takes to train the model. For general users, a pre-trained model with learning across multiple tables might be beneficial.”
>
> **R:** You’re right, a pretrained model which can learn across datasets is definitely something we are interested in and plan to explore in the future. We allude to this in our future work section.
>
> **L:** ”The presented work might not be accessible to regular users, given the amount of resource required to train the model.”
>
> **R:** The compute used by our model is more than some of the other models we compared against, but we do still believe it should be accessible for most users to train for most use cases. Training on a single GPU takes around 0.5-2.5 hours on most datasets. A strong model for our scaling dataset with tens of millions of rows can be trained within a GPU day. These models as a whole use much less compute than ones used in NLP or Vision, but we still felt it important to mention the compute usage as a limitation, since it is often ignored.
>
> **Q:** ”Is it possible for the model to generate new categories (or numbers) not present in the train set?”
>
> **A:** Generating new categories is very difficult to do, at the time of writing, we are not aware of any model which can do this well in general. However for ordered or continuous variables our quantization means most values generated are not present in the training set. Additionally, altering the quantizer to be a GMM instead of the default K-Means, or interpolating between the support allows us to generate arbitrary values. Our default model does not do this, since we found the simpler finite support method still gave us strong results overall. The ability of our model to condition on any subset of fields during generation means it can also be seamlessly combined with other methods of generating data, for example, to create a joint diffusion-masked model. Prior work is unable to do this.
>
> Again, thank you very much for your time and review. We are glad you liked the paper, and we hope it can help researchers and practitioners to examine new research directions and solve more problems.

---

### Official Review · Reviewer_gJJU · 2023-07-07

**Soundness:** 3 good
**Presentation:** 3 good
**Contribution:** 3 good
**Rating:** 4
**Confidence:** 4

**Summary:**

This paper proposes a new generative model of table-type data based on the transformer.  And it can address the unique challenges posed by heterologous data fields and natively handle missing data.


**Strengths:**

1. TabMT is a simple but effective Masked Transformer design for generating tabular data.
2. We highlight the applicability of our model in privacy-focused applications, illustrating TabMTs ability to arbitrarily trade-off privacy and quality through temperature scaling.
3. Experiments show the effectiveness of the proposed method.

**Weaknesses:**

1. The TabMT structure has only been modified in terms of the input layer, without improving the multi-layer transformer structure. In other words, this paper only made adaptive improvements to the data format and is not on par with NeurIPS.
2. The application method of the model structure proposed by TabMT is not clear. Categorical and Numerical use independent inputs, which makes it necessary to determine the type of data input into the model. How to accurately and automatically identify this type is challenging, which also limits the complexity of generating table data for the model, making it difficult to cope with complex scenarios. The effectiveness of the model should be evaluated for inaccurate category recognition in the experiment.

**Questions:**

See weaknesses.

**Limitations:**

Yes.

---

> ### Author Rebuttal · Authors · 2023-08-08
>
> Hi Reviewer gJJU, Thank your for taking the time to read and review our paper.
>
> **W:** The TabMT structure has only been modified in terms of the input layer, without improving the multi-layer transformer structure. In other words, this paper only made adaptive improvements to the data format and is not on par with NeurIPS.
>
> **R:** We have contributed more than changing the input layer within our paper. Our paper’s goal is not to alter the transformer structure as a whole. We construct a Tabular Data Generator which achieves **state of the art quality** using a model design **fundamentally different from existing models**. Specifically we do so using Masked Transformers, which are very understudied for generation purposes and we show they suit the problem of tabular data generation very well.
>
> We contribute the following:
>
> - We construct a Masked Transformer architecture and training task for the tabular domain.
> - We improve upon the traditional masking task itself, and outline a procedure to generate data from this new model. (Masked Transformers are typically poor data generators)
> - We introduce ordered embeddings to help our model deal with numerical values better.
> - We thoroughly justify our architecture, intuitively, mathematically, and empirically.
> - We achieve State of the Art Generation Quality on over a dozen benchmarks.
> - We achieve State of the Art Privacy at these generation qualities, while being able to tradeoff privacy and quality.  Most other models can't tradeoff between these and are fixed .
> - We show how our architecture changes allow it to deal with missing data natively, something existing models can’t do.
> - We explain how our model can be used to condition on arbitrary subsets of data, in a way that other generators cannot.
> - We scale our model to tens of millions of rows and show it still performs well, a much larger scale than previous works.
> - To our knowledge, this is the most thoroughly evaluated and highest performing transformer for generating tabular data. It is also the highest performing model across all existing generative tabular model families including GANs, VAEs, Diffusion Models, and Autoregressive Transformers while using a completely different and novel generation scheme.
>
> These points are all outlined in the paper. While we have contributed much more than just changing the input layer, if this change alone achieved state of the art, while having the unique privacy tradeoff, conditioning, and missing data capabilities our model has, we believe this would be notable and something to pay attention to. **We are not aware of another model which even has two of these capabilities**, certainly not while also achieving SoTA generation quality.
>
> **W:** The application method of the model structure proposed by TabMT is not clear. Categorical and Numerical use independent inputs, which makes it necessary to determine the type of data input into the model. How to accurately and automatically identify this type is challenging, which also limits the complexity of generating table data for the model, making it difficult to cope with complex scenarios. The effectiveness of the model should be evaluated for inaccurate category recognition in the experiment.
>
> **R:** We do not aim to automatically learn or identify which columns are categorical and which columns are numerical. We assume that this is known ahead of time, and that the model is allowed to treat categorical and numerical variables differently within the model. In the majority of cases, the data type (float, integer, string) is sufficient for this purpose, but it should be known metadata by the practitioner. As far as we are aware, this assumption is used by essentially all tabular generators.
>
> **References:** \
> TVAE and CTGAN: https://arxiv.org/pdf/1907.00503.pdf
>
> TabDDPM: https://arxiv.org/pdf/2209.15421v1.pdf
>
> CTabGAN+: https://arxiv.org/pdf/2204.00401.pdf
>
> Again, Thank you very much for reading and reviewing our paper. We hope our responses can help clarify some things and explain why we believe our work is an impactful and novel contribution to the area of Tabular Data Synthesis.

---

### Official Review · Reviewer_sX8C · 2023-07-07

**Soundness:** 3 good
**Presentation:** 2 fair
**Contribution:** 3 good
**Rating:** 6
**Confidence:** 3

**Summary:**

This paper explores the effectiveness of masked transformers as generative models for synthetic tabular data generation. The proposed TabMT architecture effectively handles challenges related to heterogeneous data fields and missing data. The model shows promising experimental performance and demonstrates good performance even under privacy constraints.

**Strengths:**

- The paper proposes a promising masked transformer approach TabMT for generating tabular data. This is an elegant application of masked transformers.
- The paper provides comprehensive experimental evaluation and compares to a diverse set of baselines on 15 datasets
- The paper explores privacy-preserving capabilities of TabMT and shows promising results

**Weaknesses:**

- While the paper provides adequate experimental evaluation, it would be helpful to include recent state-of-the-art generative methods in evaluation such as STaSy [1]
- In the privacy experiments, only TabDDPM was used as the baseline, it would be interesting to see more comparisons with other baselines

[1] Kim, J., Lee, C. and Park, N., 2022. Stasy: Score-based tabular data synthesis. arXiv preprint arXiv:2210.04018.

**Questions:**

1. Could you please clarify the training procedure? When training the masked transformer, does every feature have its own classification head for predicting the mask?
2. Since you explore hyperparameter tuning for the downstream Catboost model, how was hyperparameter tuning performed for each of the generative models?
3. In the privacy experiments, only TabDDPM was used as the baseline, would it be possible to add more comparisons with other baselines?
4. It would be great to add recent state-of-the-art generative models into evaluation, such as STaSy [1]

[1] Kim, J., Lee, C. and Park, N., 2022. Stasy: Score-based tabular data synthesis. arXiv preprint arXiv:2210.04018.

**Limitations:**

Addressed

---

> ### Author Rebuttal · Authors · 2023-08-08
>
> Hi Reviewer sX8C, thank you for your careful consideration and review of our paper.
>
> **Q:** Could you please clarify the training procedure? When training the masked transformer, does every feature have its own classification head for predicting the mask?
>
> **A:** When predicting we use a separate linear layer for each feature, but there are no extra transformer blocks or any other separate layers. It is equivalent to using a single linear layer for all features and masking out impossible values before prediction. The parameters are shared with the embedding layers.
>
>
> **Q:** Since you explore hyperparameter tuning for the downstream Catboost model, how was hyperparameter tuning performed for each of the generative models?
>
> **A:** Each dataset has a fixed set of catboost hyperparameters which are used for all evaluations and all methods on that dataset. The hyperparameters are found by tuning them on the real dataset to maximize the validation score. We use the same tuned values as other works to ensure a fair comparison. They were found using 100 tuning trials with 5 hyperparameters of Catboost.
>
> **Q:** In the privacy experiments, only TabDDPM was used as the baseline, would it be possible to add more comparisons with other baselines?
>
> **A:** We chose to evaluate against TabDDPM because its quality scores were closest to ours, creating a stronger comparison. Privacy and quality tradeoff with each other, so comparing privacies at vastly different qualities is less useful. However, here are some privacy scores from CTABGAN+:
> | Method       | AD           | CA           | CAR          | CH           | DI           | KI           |
> |--------------|--------------|--------------|--------------|--------------|--------------|--------------|
> | **TabMT (ours)** | **1.01(0.811)**  | **0.117(0.832)** | **0.041(0.737)** | **0.281(0.758)** | **0.243(0.740)** | **0.335(0.868)** |
> | CtabGAN+     | 0.119(0.772) | 0.056(0.525) | 0.012(0.733) | 0.212(0.702) | 0.196(0.734) | 0.226(0.444) |
>
> We obtain both higher privacy and quality on all datasets here.
> We can add the full results for this model to the paper.
>
> **Q:** It would be great to add recent state-of-the-art generative models into evaluation, such as STaSy
>
> **A:** We are happy to mention StaSy and score-based modeling in our related work section. We have looked over StaSy and it is certainly a strong paper. Unfortunately, none of the datasets the paper tests against overlap with the ones we test with. Implementing StaSy and training it across our fifteen datasets would be quite time-consuming. Additionally, training our model on StaSy’s datasets would also take a fair bit of effort and would be disconnected from the other evaluations we perform. Due to the considerable effort involved, we likely won’t be able to include it in the evaluations at this time.
>
>
> Again, thank you very much for your response to our paper and the time you took to review it.

---

> > ### Comment · Reviewer_sX8C · 2023-08-21
> > **Response to the author rebuttal**
> >
> > I thank the authors for their response and provided clarifications. Regarding hyperparameter tuning, my question was about the hyperpameter tuning of the generative models rather than catboost. How was hyperparameter selection performed for both TabDDPM and the baseline generative models it was compared to?
> >
> > Regarding implementing StaSy, in fact it’s official GitHub provides a user-friendly implementation: https://github.com/JayoungKim408/STaSy making experimentation on at least a few datasets feasible within the rebuttal period. As StaSy is a recent and strong generative tabular model, it is important to include it in comparison with TabDDPM.

---

> > > ### Author Response · Authors · 2023-08-21
> > > **Response to Review Comment**
> > >
> > > Hi Reviewer sX8C,
> > > Thank you for your response.
> > >
> > > Hyperparameter tuning details and search space for TabMT are available in the supplementary material. We use 50 trials, just as prior work has. For the baselines, we used the same search spaces as previous work.  This ensures a fair comparison. The cited baseline work have these search spaces available.  Our reported metrics for these techniques match with metrics reported in prior work. The cited TabDDPM baseline work has a good summary of these search parameters.  If you like, we can include the TabDDPM summary in the Appendix of our paper.
> > >
> > > The deadline for the rebuttals was mere hours after your response, and therefore including results for STaSY before the deadline is infeasible. An earlier reply would have made this a feasible task. It takes days to tune baselines properly on each dataset to ensure correctness and to guarantee an accurate comparison. We compare against 4 other state of the art techniques across 15 datasets, each representing the best we could find at the time across major generative modeling families (Diffusion, GAN, Autoregressive, VAE). This comparison alone is more than necessary for a publication and a contribution to the scientific literature.  Our work is of a new masked generative model, different from these existing ones, which outperforms all tested baselines using this novel method. We also show SoTA performance and more evaluation in our scaling experiments. It's not possible to compare against every paper on ArXiV, especially without any dataset overlap. The STaSY paper was not accepted in a conference until late February, at which point we were well past looking for additional baselines to compare against.
> > >
> > > The StaSY paper appears to be a strong tabular data generator.  But we cannot faithfully (or ethically) represent the STaSY paper in our paper until we have fully evaluated the quality, repeatability, and accuracy of the STaSY method.
> > >
> > > Again, Thank you for your time.

---

> > > > ### Comment · Reviewer_sX8C · 2023-08-21
> > > > **Final Response**
> > > >
> > > > In light of the author response and lack of experiments comparing to an important relevant baseline StaSy (https://openreview.net/forum?id=1mNssCWt_v accepted to ICLR in January 2023  as a top 25% paper, and with a user-friendly implementation openly available https://github.com/JayoungKim408/STaSy), I will not be increasing my score.

---

> > > > > ### Author Response · Authors · 2023-08-21
> > > > > **Author Response**
> > > > >
> > > > > Hi Reviewer,
> > > > >
> > > > > We are sorry to hear that, but we appreciate the time you've taken on this. The request with a link to the code was not made until hours before the deadline. You are correct it was published in early February not late, sorry that was a misreading. But, the code was not linked on that post until late May. A googling of "stasy tabular score based modeling github" did not return the codebase for this paper. We have compared against many strong baselines, and it is unclear to us why this particular baseline would be this critical. The reviewer has previously stated they believe our evaluation to be strong.
> > > > >
> > > > > Thank you.

---

> > > > > > ### Comment · Reviewer_sX8C · 2023-08-21
> > > > > > **A comment for AC**
> > > > > >
> > > > > > Dear Area Chair,
> > > > > >
> > > > > > I would like to emphasize that I linked the *official implementation* GitHub of the StaSy paper available on the openreview with ICLR acceptance decision for that paper. That is, I provided enough information in my original review by citing the paper itself. I would expect the authors to be able to find the implementation themselves without asking the reviewers for links. Again, this is an important baseline accepted as a top-25% ICLR paper on *Jan 20, 2023* — well before the NeurIPS submission deadline. In addition to the GitHub link, the code was available as a supplementary material for the StaSy paper.
> > > > > >
> > > > > > Thank you.

---

> > > > > > > ### Author Response · Authors · 2023-08-21
> > > > > > > **A comment for AC**
> > > > > > >
> > > > > > > Dear Area Chair,
> > > > > > >
> > > > > > > Including accurate results from another method takes many weeks.  Our paper compares 4 methods, across 15 datasets, with over 50 baseline metrics from state of the art published papers within the last year, to include the last several months.  We do not think that another comparison would significantly enhance or degrade the results or impact of our paper.  We are sorry we did not find the implementation on GitHub, but again we did not have enough time between the rebuttal last week and the reviewer response this morning either way. If last week the reviewer had said they would still require that we include these results, then we could have attempted to include some preliminary results then. A proper assessment requires time to run the necessary tests, analyze the results, and then include them in the paper.
> > > > > > >
> > > > > > > Respectfully,
> > > > > > > The Authors

---

### Comment · Area_Chair_pdFu · 2023-08-21
**Mixed reviews**

Dear reviewers,

Thanks for the hard work so far!

This paper received mixed reviews and the authors have responded each of you.

We need to ideally reach a consensus in the rebuttal period, and at least should have active discussions, updated recommendations, and acknowledgment that you have read the response.

Can you please check other reviews and the author rebuttal and see if your opinion has changed? Please give your reasoning in as much detail as possible.

AC

---

### Decision · Program_Chairs · 2023-09-21

**Decision:**

Accept (poster)

**Comment:**

The paper received accept, weak accept, and two borderline rejects post rebuttal. However, 3 out of the 4 reviewers (except the weak accept one sX8C) did not update their recommendation. The AC checked all the materials, including the author response to each of the reviewers, and the supplementary materials, and decides to accept the paper. Overall, the paper extends the research of tabular data generation with masked Transformers, and show extensive comparisons with prior works justifying the proposed design. Please incorporate the feedbacks from the reviews for the final version.